# Cyber Secure Framework for Smart Agriculture: Robust and Tamper-Resistant Authentication Scheme for IoT Devices

**Saleh Alyahya** [1,†] [ID]**, Waseem Ullah Khan** [2,*,†] [ID]**, Salman Ahmed** [2,†] [ID]**, Safdar Nawaz Khan Marwat** [2,†] [ID]
**and Shabana Habib** [3,†]

1   Department of Electrical Engineering, College of Engineering and Information Technology, Onaizah Colleges, Unaizah 56447, Saudi Arabia; saleh.alyahya@oc.edu.sa
2   Department of Computer Systems Engineering, University of Engineering and Technology, Peshawar 25120, Pakistan; sahmed@uetpeshawar.edu.pk (S.A.); safdar@uetpeshawar.edu.pk (S.N.K.M.)
3   Department of Information Technology, College of Computer, Qassim University, Buraidah 51452, Saudi Arabia; s.habibullah@qu.edu.sa
*   Correspondence: waseem@uetpeshawar.edu.pk
†   These authors contributed equally to this work.

**Abstract:** Internet of Things (IoT) as refers to a network of devices that have the ability to connect, collect and exchange data with other devices over the Internet. IoT is a revolutionary technology that have tremendous applications in numerous fields of engineering and sciences such as logistics, healthcare, traffic, oil and gas industries and agriculture. In agriculture field, the farmer still used conventional agriculture methods resulting in low crop and fruit yields. The integration of IoT in conventional agriculture methods has led to significant developments in agriculture field. Different sensors and IoT devices are providing services to automate agriculture precision and to monitor crop conditions. These IoT devices are deployed in agriculture environment to increase yields production by making smart farming decisions and to collect data regarding crops temperature, humidity and irrigation systems. However, the integration of IoT and smart communication technologies in agriculture environment introduces cyber security attacks and vulnerabilities. Such cyber attacks have the capability to adversely affect the countries' economies that are heavily reliant on agriculture. On the other hand, these IoT devices are resource constrained having limited memory and power capabilities and cannot be secured using conventional cyber security protocols. Therefore, designing robust and efficient secure framework for smart agriculture are required. In this paper, a Cyber Secured Framework for Smart Agriculture (CSFSA) is proposed. The proposed CSFSA presents a robust and tamper resistant authentication scheme for IoT devices using Constrained Application Protocol (CoAP) to ensure the data integrity and authenticity. The proposed CSFSA is demonstrated in Contiki NG simulation tool and greatly reduces packet size, communication overhead and power consumption. The performance of proposed CSFSA is computationally efficient and is resilient against various cyber security attacks i.e., replay attacks, Denial of Service (DoS) attacks, resource exhaustion.

**Keywords:** Internet of Things; authentication; security; agriculture

## 1. Introduction

With the recent advancements in communication and physical sciences, it is now possible to connect physical objects to the Internet from any place any time for anything without human intervention by making everything smart. This advancement is referred to as the Internet of Things (IoT) [1]. One of this century's key technological development has been the IoT, which is still in its fancy stages. In the coming years, this development is expected to hit a big figure. By the end of 2024, 62 billion IoT devices are expected to exist [2]. According to Statista report, it is predicted that this number will rise to 75.4 billion by 2025. The significance of the IoT system has been acknowledged by major international standards bodies, thereby ensuring development towards the proper functionality, flexibility and

compatibility of this system. Due to which IoT is making an impact in the majority of the fields i.e., healthcare, traffic, logistics, oil and gas industries and agriculture [3]. The IoT enables devices with a variety of features and capabilities to connect to the Internet. The rapid rise of IoT applications results to a tremendous increase in the number of IoT devices connected to the worldwide network as well as network traffic over conventional network. Due to its constrained nature, the IoT growth adds more security challenges to the conventional network, as the conventional communication network already faces various security challenges. Different IoT devices are not constructed to the necessary security requirements, resulting in huge security breaches [4].

Food consciousness has grown amongst the people lately with the exponential increase of the food industries and radical improvements in individuals' dietary behaviors and ways of life [5]. People nowadays are more worried about the quality and safety of the food they consume. Food safety is perceived to include food that is liberated from contaminants and chemicals that may cause the development of microscopic organisms destructive to individuals' safety and lives [6]. Agriculture systems has a significant impact on a country's economic growth as it is the main source of food. According to report published by the United Nations' Food and Agriculture Organization [7], a significant portion of food waste happens due to various variables during the pre-harvesting, harvest and even post-harvest stages and as well the extent of its influence on the economy, environment, health and survival of human race. According to a UN report, the world population is estimated to surpass 9 billion humans being by the end of 2050, an increase of about a third over the current population [8]. More than half of this increase will occur in Pakistan, Nigeria, India, Indonesia, Ethiopia, the United Republic of Tanzania, Egypt, and the United States. Such rise in population necessitates a nearly 70 percent increase in food production rate. This fast growing population brings with it a slew of additional challenges, including increased competition and overuse of land, water and also other natural resources. These challenges highlight critical need to minimize food system's reliance on our environment. As a result, in order to meet the expanding need for food and crop production, an evolutionary agricultural paradigm is required to ensure long-term development. In an agricultural field, a smart sensing system is necessary to integrate the status of the field, its influence on field productivity, and to determine the actions that must be taken on the field for a good outcome of a good produce. A smart sensing environment is made up of a system of connected devices that can constantly send and receive data from one another. It can also make decisions on behalf of a user and take action to enhance the environment [9]. This shift in the environment necessitates more constant monitoring of the surroundings, resulting in an ever-changing environment.

Agriculture plays an important role in country's economic growth and it is the basis for human species. Conventional agricultural methods are still being used by farmers, resulting in low crop and fruit yields. As a result, integrating IoT technologies can boost crop productivity. IoT is an essential part of the smart agriculture architecture. With the integration of IoT devices, the agriculture systems have been monitored 24/7. These IoT devices sense the surrounding environment and physical parameters to take sensed information. Such sensed information is then sent to the backend server via Internet or through sensor gateway, where date is stored for future use or even user can view that sensed data. The user can make decisions about what actions to do in such a monitored environment. However, it's not an easy task to integrate everyday devices with the Internet. Security is a big challenge faced by IoT, although each device has its unique set of characteristics and requirements. First of all, each person, device and system which is connected to the Internet must be identified. Intruders will acquire access to the network and perform security breaches if they do not have a valid identity.

In case of smart agriculture systems, if information of sensors monitoring the agriculture parameters is leaked, lost or tampered with, it can render the products useless and damage their value significantly hence resulting in huge economic food loss. Therefore, developing a concrete framework for secure monitoring of smart agriculture with the adap-

tion of IoT devices is an essential requirement of this era. Since IoT devices are designed to be compact and cost effective, because they have limited computing power and memory. Their resource constrained nature makes it difficult to secure them and has necessitated the development of lightweight application protocols and security suites. Instead of using standard TCP/IP protocols i.e., Hyper Text Transfer Protocol (HTTP) and Transport Layer Security (TLS), these devices use the IoT protocol stack, which is a lightweight version of the TCP/IP model specially designed for IoT devices. The CoAP has been standardized as lightweight application protocol for IoT systems. However, developing a low-overhead security solution for CoAP remains a challenge. Though some security protocols i.e., standard Datagram Transport Layer Security protocol (DTLS) exists. However, it is not suitable for IoT constrained devices and is expensive due to its computational overhead, complexity and lengthy cipher suites process. Therefore, this paper proposed a robust and lightweight secure authentication framework in CoAP using symmetric encryption i.e., Advanced Encryption System (AES) and HMAC-SHA-224. The main contribution of this paper is to design robust and tamper-resistant secure authentication framework for agriculture application using CoAP by ensuring data integrity and authenticity. The proposed CSFSA is implemented in Contiki NG and Cooja simulator and evaluated performances in terms of packet size, communication overhead and power consumption. The significance of proposed CSFSA is to make resilient against various cyber security attacks i.e., replay attacks, Denial of Service (DoS) attacks, resource exhaustion.

The proposed CSFSA provides secure monitoring of various agriculture parameters remotely. The architecture of proposed CSFSA has three modules i.e., 6LoWPAN based internal network having IoT devices deployed in agriculture field for monitoring and controlling, a border router which acts as bridge between Internal network and external network i.e., Internet and a web server that provides access to the sensed data as shown in Figure 1.

The paper is structured as follows. Section 2 provides brief explanation of literature review that has been done in smart agriculture, IoT and CoAP security. Section 3 provides an in-depth overview of standards and protocols pertinent to our research work. Proposed methodology is presented in Section 4. Simulation and results evaluation is discussed in Section 5. Finally, concluding remarks are given in Section 6.

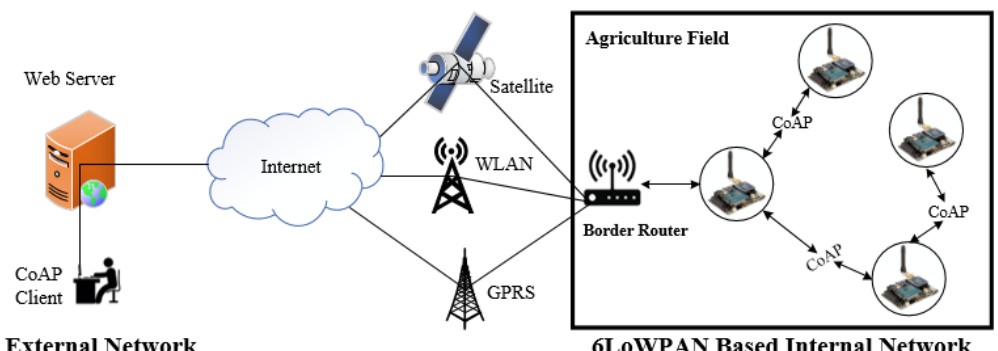

**Figure 1.** CSFSA architecture.

## 2. Related Work

This section provides the related literature that has been done in the field of CoAP security, IoT security, various authentication techniques used between IoT devices and agriculture sector.

The Hyper Text Transfer Protocol (HTTP) is an application layer protocol used by the World Wide Web to access data on Internet and uses the Representational State Transfer (REST) architecture. The HTTP request and response messages exchange data between HTTP server and client [10]. This protocol is not suitable for IoT and cannot be implemented

on IoT devices due to computational overhead. As a result, a lightweight and efficient protocol for constrained devices is required. The Internet Engineering Task Force (IETF) Constrained RESTful Environment (CoRE) working group has standardized lightweight application protocol for IoT systems i.e., CoAP. It is a specialized web transfer protocol, specially designed for resource constrained applications. In [11], the authors compared the CoAP and HTTP protocol and evaluated their performances based on energy consumption, battery life and amount of bytes exchanged per session as shown in Table 1.

**Table 1.** HTTP and CoAP comparison based on resource consumption.

| Protocol | Bytes per Transaction | Power | Life Time |
|:---:|:---:|:---:|:---:|
| HTTP | 1451 | 1.333 mW | 84 days |
| CoAP | 154 | 0.744 mW | 151 days |

Transport Layer Security (TLS) is used for securing HTTP communication over Transmission Control Protocol (TCP). The CoAP communication over User Datagram Protocol (UDP) is secured by using DTLS. However standard DTLS is not suitable for constrained networks due to its computational overhead, complexity and handshake session overhead. In [12], the authors provided brief overview of layered architecture of IoT and security vulnerabilities at each layer. They discussed in details various communication technologies along with their features and limitations which are being used by IoT applications. They also reviewed the existing security mechanisms with their limitations to secure the IoT environment. In [13], the authors presented two-way authentication scheme based on DTLS and RSA cryptographic algorithm. DTLS protocol is implemented in context of system architecture to achieve low overhead and high interoperability. However, due to the deployment of RSA encryption, the computational overhead required by their handshake process requires more energy. Shahid et al. [14] presented lightweight secure CoAP i.e., Lithe for securing IoT devices communication. This work provided secure framework by integrating CoAP and DTLS. The standard DTLS is not suitable for constrained devices. The proposed scheme used the DTLS header compression by leveraging 6LoWPAN standard and significantly reduced the energy consumption. The results showed that compressed DTLS performs better in terms of energy efficiency, network response time, packet size and processing time.

In [15], the proposed framework enables the communication between android phones using DTLS secured CoAP. This framework combines other algorithms with DTLS that are not applicable on resource constrained IoT devices. DTLS requires public key cryptography which is a computationally expensive process and a lengthy handshake process [16]. Although there are certain security algorithms which are lightweight and have less overhead such as raw public key, at least 25 bytes per-packet overhead of DTLS takes up one-third of a frame. DTLS heavily hinders the performance in terms of round-trip messages when retrofitted to CoAP [17].

The authors Raza et al. [18], present a scheme for making the implementation secure CoAP lightweight by applying header compression techniques. Although header compression does not offer an improvement in performance because DTLS is computationally intensive. The authors Freeman et al. [19] propose a plan that enables the client to assign a certificate path and construction as well as the validation of the certification path to a server. However, this protocol is expected to be used over HTTP. This protocol relies on third-party instead of communication parties and adds a heavy communication overhead. In [13], a framework for the end-to-end security of IoT has been presented. The authors have used DTLS in combination with other standard protocols such as RSA. The use of RSA adds a high computational complexity and overhead to the network. In addition to that, the authentication handshake process in DTLS in which the client and server authenticate each other takes in excess of three round trips. Authors of [20] have suggested a technique for lightweight authentication using Advanced Encryption Standard (AES). In this tech-

nique, the client and server ensure the identity of the devices based on the message payload. During this process, a total of four messages are exchanged between the client and server.

In [21] Park et al. have presented an implementation of an architecture that uses a delegation method to send and receive messages. In this implementation, a Secure Service Manager (SSM) is used in the handshake process. This system requires the SSM, sensors and trust manager to be located in a constrained network. Hence it may not be suitable for all kinds of IoT setups. In [22], Granjal et al. propose a plan for end-to-end security by using DTLS algorithm. This system uses the Elliptic Curve Cryptography (ECC) to encrypt data in resource constrained nodes. Although ECC consumes less power and memory compared to other public key encryption algorithms, it still uses up more resources than the symmetric key encryption algorithms [23]. Using complex cipher suites for encryption results in high energy consumption. The authors have used a symmetric key cryptography algorithm, AES to replace DTLS in the authentication process. AES performs a four-way handshake between the CoAP based server and CoAP client. This process adds two header options in the CoAP packet for enabling secure mode.

In [24], the authors present a scheme for the authentication between client and server by modifying the header of the CoAP packets. AES 128 is used for the encryption process and UDP protocol is used without DTLS in the transport layer. This scheme reduces the number of messages required for authentication between two parties without any loss in the overall efficiency. In [25], the various advantages of the new features of "Future Internet" are described, which greatly benefits farm management techniques. The "Future Internet" programme overcomes the limitations of the already existing Internet. These benefits include generic software modules that can be modified to build specific modules related to farming. The main architecture of the farm management system is specified, depending on which model is stored in the cloud.

Tanmay Baranwal et al. [26] the main focus of this paper is to protect and secure agricultural goods from rodents and insects in storage units and fields. Real-time security systems are used, which provide notifications as soon as an issue is identified. Python scripts are used to integrate the various electronic devices and sensors. The algorithm of this system is based on data collection to achieve accuracy in alerting the user and starting the repellent. The system is tested in a 10 m², the device is fixed in a corner during the testing. As the PIR sensor senses heat, it immediately starts the URD sensor and starts video recording through a webcam. The accuracy achieved from these test cases reaches up to 84.8%. It will be beneficial to incorporate the security system in storage units to prevent rodents from attacking the goods. LIU Dan et al. [27] this paper emphasizes on the role of greenhouse technology in the agricultural field and presents the design and development of an environment monitoring system. This system makes use of ZigBee technology and CC2530 chip. Data collection, processing and communication by the wireless sensors and nodes is done by using the CC2530F256 core. The proposed monitoring system gives real-time information to the user regarding different parameters such as temperature control and fan condition etc. This system utilizes intelligent greenhouse monitoring and control techniques and helps the farmers to produce balanced and scientific crops.

Nikesh Gondchawar et al. [28] proposes a smart agriculture system based on IoT and automation techniques. The proposed system uses a remotely controllable robot based on GPS to execute tasks such as weeding, spraying pesticides and monitoring moisture levels etc. The proposed system also incorporates smart irrigation techniques through intelligent decision making based on real-time data from the fields. In addition to that, it also includes smart warehouse management modules. It observes the temperature, humidity as well as theft detection in the warehouse. The system architecture includes various sensors, cameras, ZigBee devices, actuators with microcontrollers and Raspberry Pi. This system alerts the user regarding irrigation and storage problems and uses a remotely controllable robot for smart irrigation and warehouse management.

Agostino Forestiero [29] proposes an activity footprints based method to detect anomalies in IoT by exploiting a multiagent algorithm. The method allows real-valued vectors to

be used to map sequences of specific device activity (digital footprints). The vectors are given to mobile agents, who act according to a modified bio-inspired model for each of them. On the basis of simple local movement rules observed by all agents onto a virtual 2D space, this model allows intelligent global behaviour to evolve. In [30], the authors proposes an agent-based algorithm for achieving a distributed resources organization in an IoT environment. In order to map the IoT objects, a natural language processing approach was used, which was able to capture the semantic context and represent things with high-dimensional vectors, allowing advanced agents to operate. In a highly dynamic and, essentially, unstructured environment, the growing organised virtual structure, i.e., a similarity-based overlay network of agents, allows for the development of informed resource selection/discovery services, making them more efficient. The approach's validity has been confirmed by preliminary results.

Marco Lombardi et al. [31] proposed a preliminary study case of a probabilistic approach in an intrusion detection system over the CAN-bus to guarantee cyber security inside connected vehicles. An innovative two-step detection algorithm has been used to exploit both the variation of the status parameters of the various ECUs over time and the Bayesian networks which can identify a possible attack. Starting from a domain analysis is possible to find out what are the parameters of interests and how these are related to each other. Marco Lombardi et al. [3] provided brief overview of IoT regarding current architectures, technologies, protocols, and applications. They discussed in details various applications which is based on IoT i.e., smart Cities, smart roads, and smart industries.

Exiting security protocols i.e., standard Datagram Transport Layer Security protocol (DTLS) is not suitable for IoT constrained devices and is expensive due to its computational overhead, complexity and lengthy cipher suites process. However, the DTLS protocol needs to be modified in such a way that it's feasible for resource constrained devices. Hence, researchers are looking for alternative security frameworks to replace DTLS. In this research work, we propose a lightweight and robust authentication framework for agriculture application and evaluate its performance to show that it is feasible for constrained IoT devices.

## 3. Background

This section provides an in-depth description of IoT layered architecture, CoAP architecture, proposed framework architecture, standard and protocols related to our research work.

### 3.1. Iot Protocol Stack Development

The IoT protocol stack is a critical component of IoT technology because it allows hardware to communicate data in a structured and useful way. In recent years, the IoT has seen an uptick in a variety of applications, such as smart homes, smart health, smart agriculture, and smart logistics, where each application has its own properties and resource requirements, as well as its own protocols to match its objectives [32].

IoT protocols are classified based on the role they play in the network. Table 2 shows the logical layering of protocol hierarchies used in network communications nowadays. The following is a comparison of the protocols for each layer in the protocol stack to determine which one is best for the smart agriculture scenario in terms of efficiency and productivity.

**Table 2.** IoT Communication Protocols.

| | |
|---|---|
| Application Layer | CoAP, MQTT, MQTT-SN and XMPP |
| Network Layer | 6LoWPAN, ZigBee and BLE |
| Link Layer | IEEE 802.15.4, 802.11 a/b/g/n/ad/ac and 802.15.1 |

### 3.1.1. Application Layer Protocols Comparison

CoAP, MQTT, MQTT-SN, and XMPP are the most often used application layer protocols in IoT. The following paragraphs offer a comparative review to choose an ideal application layer protocol for smart agriculture based on their advantages and limitations.

Constrained Application protocol (CoAP) is a web transfer protocol that is designed to be a lightweight version of HTTP based on Representational State Transfer (REST). CoAP runs over UDP, eliminating most of the TCP overhead of HTTP, decreasing the bandwidth requirements, providing more simplicity, and makes it more appropriate for smart agriculture applications. CoAP is built upon the request/response model and supports methods such as GET, PUT, POST, and DELETE as well as unicast and multicast transmission. CoAP is efficient in terms of infrastructure, bandwidth and power usage [33].

Message Queuing Telemetry Transport (MQTT) is another protocol designed for resource constraint devices by IBM. It is based on an asynchronous publish/subscribe protocol that runs over TCP. The message pattern in MQTT includes a broker, publisher and subscriber. The broker is tasked with controlling and distributing of data packets between subscribers and publishers. The addition of TCP makes it much more reliable, however it adds more latency and causes a higher bandwidth and power consumption. Hence, it is not recommended for real time applications.

MQTT-SN is a modified version of MQTT, designed especially for sensor networks. It uses UDP instead of TCP to minimize the drawbacks of MQTT but its core infrastructure is the same as MQTT. The main difference between the two is that, MQTT-SN requires a gateway to translate all MQTT-SN messages over UDP to MQTT messages over TCP. Currently, this functionality is integrated within the brokers [34]. This extra step for the interpretation of messages, adds complexity to the overall framework hence increases power consumption.

The Extensible Messaging and Presence Protocol (XMPP) is designed to allow short messages and low latency, making it ideal for IoT communications. The XMPP protocol may handle both request/response and publish/subscribe formats, which allow for bidirectional and multidirectional communication, respectively. The decentralised nature of XMPP allows for high scalability [35]. The XMPP protocol use XML for communication, which increases network traffic. This results in high bandwidth use, high CPU usage, and no guarantee of QoS.

The comparative analysis of various application layer protocols show that CoAP generates less overhead than the other protocols. In terms of power consumption, bandwidth and infrastructure requirements and is most suitable for the proposed CSFSA framework.

### 3.1.2. Network Layer Protocols Comparison

Some of the most used network layer protocols used in IoT applications are IPv6 over Low-Power Wireless Personal Area Network (6LoWPAN), ZigBee and Bluetooth Low Energy (BLE). A comparative review of these network layer protocol is given as follows:

6LoWPAN is based on IP version 6 (IPv6), it features 128-bit hexadecimal addresses, and uses the 802.15.4 radio frequency. It enables the use of IP in low-power and lossy wireless networks, such as WSNs (Wireless Sensor Networks), IoT (Internet of Things) and M2M applications. One of the most unique characteristics of 6LoWPAN is that it allows header compression and encapsulation, making it more lightweight and secure than ZigBee. The most notable advantage of 6LoWPAN is that it supports IP networks natively [36].

The ZigBee protocol is based on the IEEE 802.15.4 standard, which allows for less battery consumption in IoT networks by keeping nodes in low power mode for the most part. However, unlike 6LoWPAN, which has interoperability, ZigBee cannot readily communicate with other protocols.

Bluetooth 5 is the latest version of BLE. It supports IP networks however, BLE is unable to establish a self-healing mesh network, which is becoming particularly crucial for IoT applications. An additional gateway is required to communicate with the internet when utilizing ZigBee or classic Bluetooth, which adds extra overhead.

It can be deduced from the above discussion that due to its compression and encapsulation capabilities, interoperability and low network overhead, 6LoWPAN is the prime option for the proposed CSFSA framework.

### 3.1.3. Link Layer Protocols Comparison

There are three main link layer protocols present for IoT applications, IEEE 802.15.4 for 6LoWPAN and ZigBee, IEEE 802.11 for Wi-Fi. and IEEE 802.15.1 for Bluetooth. A brief analysis of the above mentioned protocols is given below:

IEEE 802.15.4 is a low-power wireless network standard developed by the Institute of Electrical and Electronics Engineers (IEEE). It has a communication range of at least 10 metres and up to 100 metres. The 802.15.4 category is the dominant standard for low data-rate and security in Wireless Personal Area Networks (WPAN).

IEEE standards for wireless communications WLANs/Wireless Fidelity (Wi-Fi) are 802.11 networks with communication ranges varying from 40 to 90 metres. Wi-Fi (IEEE 802.11) is inefficient in terms of battery life, does not cover a vast area, and does not support a large number of end devices.

Bluetooth, i.e., 802.15.1, is another WPANs standard specified by the 802.15 group, with a communication range of 10 metres to 50 metres. It is more power efficient but less encrypted compared to Wi-Fi [37].

From the comparison it is determined that IEEE 802.15.4 is the best fit for the proposed CSFSA framework because it is compatible with 6LoWPAN, provides more security, consumes less power and has a higher communication range compared to the other protocols.

Based on the comparison of several protocols at different layers, the IoT protocol stack for proposed CSFSA framework is presented in Figure 2.

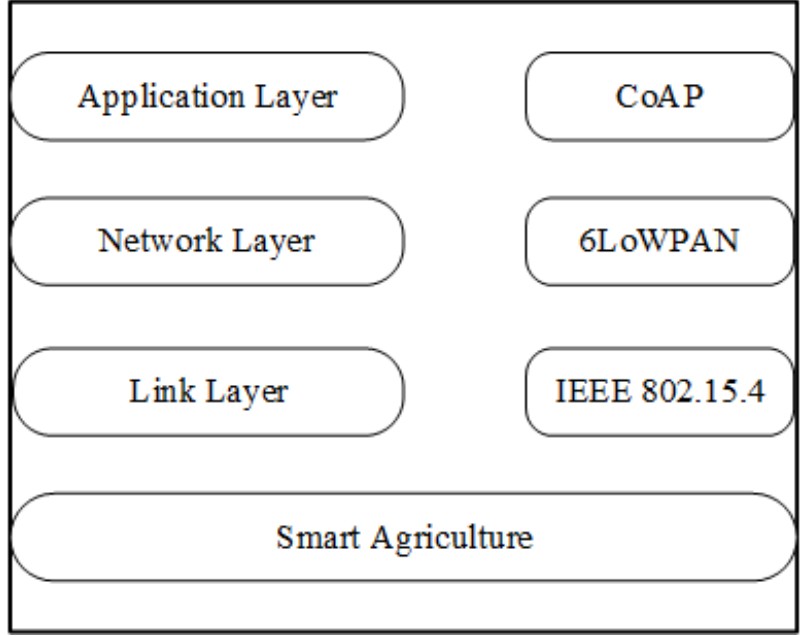

**Figure 2.** IoT protocol stack for CSFSA.

### 3.2. Coap Architecture

The Constrained Application Protocol (CoAP) is a specialized web transfer protocol, specially designed for constrained applications. The CoAP's interactive client/server model is similar to HTTP model. It has similar features and principles as in HTTP. Due to higher communication overhead of HTTP and HTTPS, such protocols are not suitable for IoT communication. Therefore, CoAP is designed to interface with HTTP, to easily integrate with existing web-browsers, while ensuring constrained networks and M2M [38,39] specialized requirements i.e., multicast support, simplicity, and low message overhead. The IETF

Constrained RESTful Environment (CoRE) working group standardized CoAP protocol. It is designed according to the Representational State Transfer (REST) architecture. CoAP supported multicast requests, asynchronous communication, content type and congestion control, cashing and proxy abilities. The CoAP packet format has a 4 bytes fixed size header as shown in Figure 3.

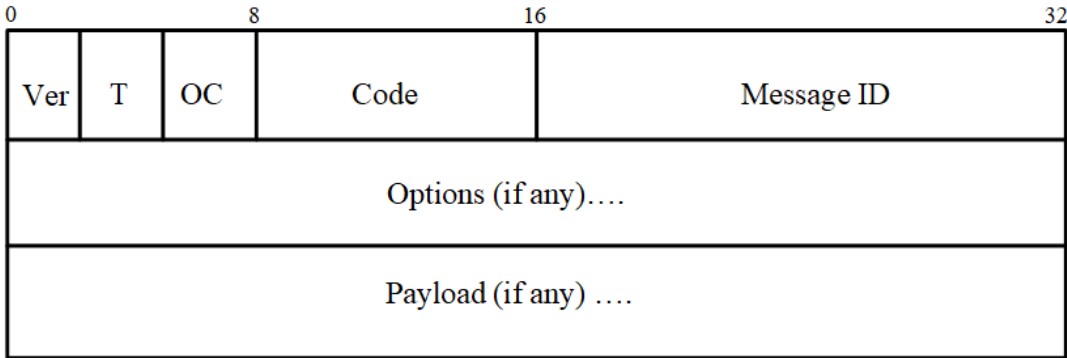

**Figure 3.** CoAP packet format.

The CoAP architecture has message layer and request/response layer as shown in Figure 4. The message layer handles the message exchange over UDP between end points. It deals with reliability and asynchronous mechanism. While request/response layer deals with request and response messages and their mapping. CoAP has four types of messages i.e., Confirmable (requires an acknowledgement), Non-Confirmable (needed no acknowledgement), Acknowledgement (receipt of confirmable message), Reset (confirmable or non-confirmable message not processed properly), Piggy-backed (sends Acknowledgement with confirmable message). The CoAP used same methods i.e., GET, POST, PUT, DELETE for generating, restoring, creating or updating and deleting processes as like HTTP. The CoAP protocol lacks trusted standards for secure architecture and its messages are encrypted by using DTLS, which was not designed for resource constrained devices and hence is not suited for CoAP [40].

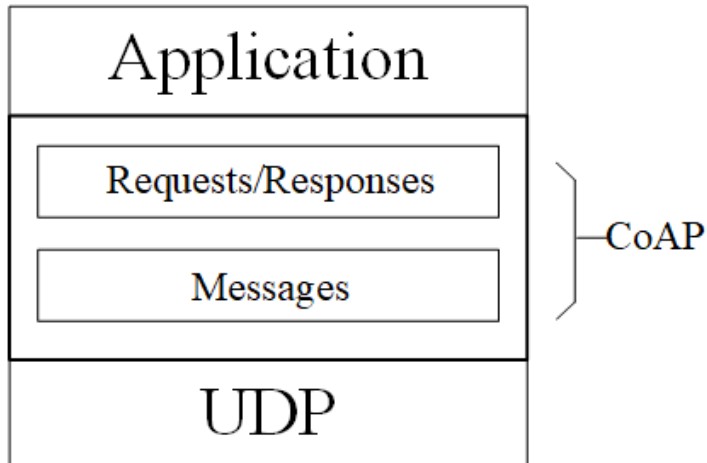

**Figure 4.** CoAP architecture.

### 3.3. Proposed Robust and Tamper-Resistant Authentication Framework

In most of the IoT applications, CoAP is utilized as an application layer protocol. CoAP does not provide any security for data transmission between two constrained nodes. The CoAP server and client communication can be secured with DTLS, however, as mentioned in the preceding section, DTLS is not suitable for IoT constrained devices and it causes computational overhead and complexity on constrained nodes. Therefore, ro-

bust and tamper-resistant authentication framework between CoAP server and client is proposed to ensure data integrity and authenticity. The proposed CSFSA framework is computationally efficient and is resilient against various cyber security attacks i.e., replay attacks, DoS attacks and resource exhaustion. It avoids the need to develop the additional security protocol.

Like HTTP, CoAP is based on client/server model architecture to establish connection when client sends request to server and server sends response to the client. For observation of resources, all the server and clients used the CoAP protocol. In order to observe resource, the server first authenticates the client identity. If the client is authentic, the server will establish session with that client and will respond to the client. On the client side while receiving response message from server, the client checks the server authenticity. Once both client and server are authenticated, they proceed and exchange data between them. The proposed CSFSA framework uses the AES encryption techniques and hash based message authentication code algorithm HMAC-SHA-224 by ensuring message integrity and authenticity. Before starting communication, the CoAP client and server first agree on sequence ID and pre-shared secret key $K_{pre}$. The server stores each device sequence ID and $K_{pre}$ in its own repository for device identification and to fetch associated $K_{pre}$ for that device. Now whenever client wants to start communication with server, it will send request to server, after receiving request from server, server will check client authenticity first. Figure 5 shows the proposed CSFSA authentication architecture. On the client side, the server verifies and authenticate the client identity. The client first generates timestamp T1 in order to prevent from replay attacks and calculates hash value of $K_{pre}$ and T1 using HMAC-SHA-224 hash algorithm. This hash value is then concatenated with the device sequence ID and T1 as follow:

$$H_C = HMAC\ (K_{pre}, T1) \tag{1}$$

$$C_{MP} = ID\ ||T1||\ H_C \tag{2}$$

where $C_{MP}$ is the client message payload. The client then sends its final message $C_{MP}$ to the server.

On the server side, while receiving message from client, the server checks client message authenticity. The server first checks the sequence ID in its repository, if it's not present in repository, the connection will be then rejected, otherwise server will fetch associated $K_{pre}$ for that device in its repository. The server then generates its own hash value of $K_{pre}$ and T1 as follow:

$$H_S = HMAC\ (K_{pre}, T1) \tag{3}$$

where $H_S$ is the server hash value. The server compares its own generated hash value $H_S$ with the received hash value $H_C$ from the client. If both hash values are not same, the server then rejects the client request. If both values are same, then the server authenticates client for services. Once client is authenticated by server, the server creates reply message for the message received from authentic client. Therefore, the server generates timestamp T2 and session key KS. The server encrypts KS and T2 using $K_{pre}$ and AES encryption algorithm and calculate the hash value of $K_{pre}$ and T2 using HMAC-SHA-224 hash algorithm as follows:

$$EM = E_{K_{pre}}\ (KS, T2) \tag{4}$$

$$H'_S = HMAC\ (K_{pre}, T2) \tag{5}$$

The server sends $H'_S$ along with encrypted message EM and T2 to the client. Upon receiving message from server, the client generates the hash value HMAC ($K_{pre}$, T2) and compares both own generated hash value and received hash value. If both hash values are not same, the client terminates the connection. If both are same, the server is authenticated

and the client then decrypts the received encrypted message EM using $K_{pre}$ and AES encryption algorithm to obtain session key KS.

$$H'_C = HMAC\ (K_{pre},\ T2) \tag{6}$$

$$DM = D_{K_{pre}}\ (EM) \tag{7}$$

In this way, the sensitive agriculture data is being encrypted and decrypted between the server and client with the help of using this session key KS.

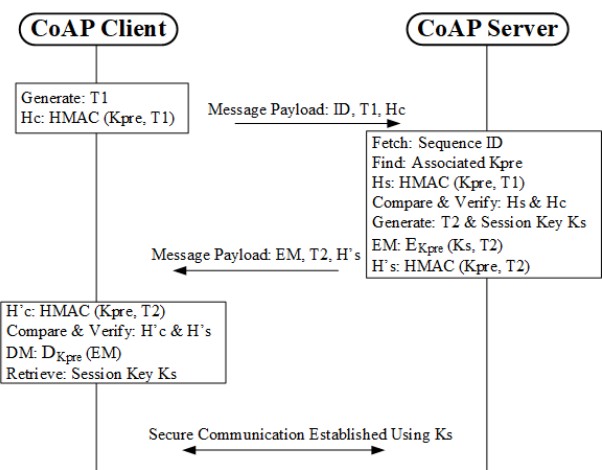

**Figure 5.** Proposed CSFSA authentication framework.

## 4. Methodology

This research work aims to develop robust and tamper-resistant authentication framework for monitoring and controlling of IoT devices in smart agriculture. The architecture of proposed CSFSA consists of three modules such as internal network i.e., 6LoWPAN based IoT network, external network and gateway which acts as bridge between internal and external network. The complete proposed CSFSA is developed and emulated in Contiki-NG operating system and Cooja simulator. Various steps of methodology are briefly explained in the following subsections.

### 4.1. Description of Proposed CSFSA Framework

This is the first phase where we will define and explain the structure of proposed CSFSA. The proposed CSFSA consists of personal computer where VMWare is installed. On VMWare, Ubuntu is installed for Contiki-NG operating system where Cooja simulator and Wireshark is used for framework development and analysis of result. Cooja is a useful simulator for Contiki-NG development, as it provides simulation environment by allowing developers to simulate and test code/system before running it on the actual hardware. The proposed framework has three modules i.e., 6LoWPAN based internal network having IoT devices deployed in agriculture field for monitoring and controlling, a gateway which acts as bridge between IoT and external network i.e., Internet and a web server.

### 4.2. Design and Development of Proposed CSFSA Framework

This is the second phase which is based on design and development. The comprehensive activity diagram for CSFSA is as shown in Figure 6. First step is to open Cooja simulation, then radio propagation model is selected i.e., Unit Disk Graph Medium-distance loss (UDGM-Distance Loss). The Unit Disk Graph Radio Medium abstracts radio transmission range as circles. Two different range parameters are used i.e., one for transmission and one for interfering with other radios and transmission.

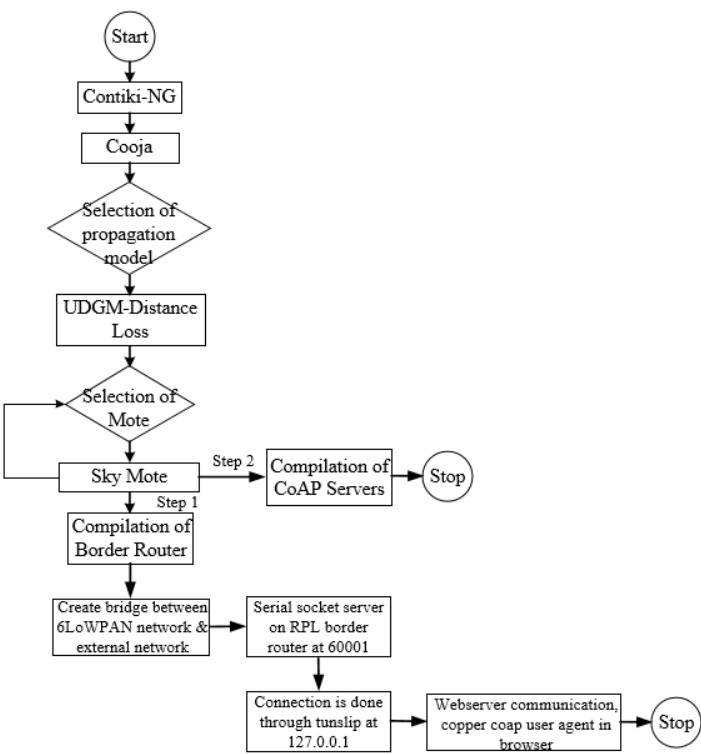

**Figure 6.** Activity diagram for design and development.

The next step is the selection of mote i.e., sky mote is selected having 10 KB of RAM. On one sky mote, CoAP server is compiled for communication within agriculture field (internal network) and is secured by deploying proposed CSFSA. While Routing Protocol for Low-Power and Lossy Networks (RPL) border router is compiled on another sky mote for connectivity with external network. The next step is to create bridge between border router and external network. To enable that bridge, need to open serial socket server on the border router through "Listening on port 60001". In this way, RPL network is created. Now to connect this RPL network to the external network, tunslip utility is used which is provided in Contiki-NG. Tunslip creates a bridge between RPL network and local machine i.e., 127.0.0.1 as shown in Figure 7.

```
user@ubuntu:~/contiki/tools$ sudo ./tunslip6 -a 127.0.0.1 aaaa::1/64
[sudo] password for user:
slip connected to ``127.0.0.1:60001''
opened tun device ``/dev/tun1''
ifconfig tun1 inet `hostname` mtu 1500 up
ifconfig tun1 add aaaa::1/64
ifconfig tun1 add fe80::0:0:0:1/64
ifconfig tun1

tun1      Link encap:UNSPEC  HWaddr 00-00-00-00-00-00-00-00-00-00-00-00-00-00-00
-00
          inet addr:127.0.1.1  P-t-P:127.0.1.1  Mask:255.255.255.255
          inet6 addr: fe80::1/64 Scope:Link
          inet6 addr: aaaa::1/64 Scope:Global
          UP POINTOPOINT RUNNING NOARP MULTICAST  MTU:1500  Metric:1
          RX packets:0 errors:0 dropped:0 overruns:0 frame:0
          TX packets:0 errors:0 dropped:0 overruns:0 carrier:0
          collisions:0 txqueuelen:500
          RX bytes:0 (0.0 B)  TX bytes:0 (0.0 B)

tunslip6: serial_to_tun: read: Success
ifconfig tun1 down
netstat -nr | awk '{ if ($2 == "tun1") print "route delete -net "$1; }' | sh
user@ubuntu:~/contiki/tools$
```

**Figure 7.** Tunslip connectivity.

To initiate border router connection, open a new terminal in Contiki and type the following commands (shown in Figure 8):

cd Contiki/examples/ipv6/rpl-border-router/
make connect-router-cooja

```
user@ubuntu:~/contiki/examples/ipv6/rpl-border-router$ make connect-router-cooja
TARGET not defined, using target 'native'
sudo ../../../tools/tunslip6 -a 127.0.0.1 fd00::1/64
slip connected to ``127.0.0.1:60001''
opened tun device ``/dev/tun0''
ifconfig tun0 inet `hostname` mtu 1500 up
ifconfig tun0 add fd00::1/64
ifconfig tun0 add fe80::0:0:0:1/64
ifconfig tun0

tun0      Link encap:UNSPEC  HWaddr 00-00-00-00-00-00-00-00-00-00-00-00-00-00-00
-00
          inet addr:127.0.1.1  P-t-P:127.0.1.1  Mask:255.255.255.255
          inet6 addr: fd00::1/64 Scope:Global
          inet6 addr: fe80::1/64 Scope:Link
          UP POINTOPOINT RUNNING NOARP MULTICAST  MTU:1500  Metric:1
          RX packets:0 errors:0 dropped:0 overruns:0 frame:0
          TX packets:0 errors:0 dropped:0 overruns:0 carrier:0
          collisions:0 txqueuelen:500
          RX bytes:0 (0.0 B)  TX bytes:0 (0.0 B)

Rime started with address 0.18.116.1.0.1.1.1
MAC 00:12:74:01:00:01:01:01 Contiki 3.0 started. Node id is set to 1.
nullsec CSMA ContikiMAC, channel check rate 8 Hz, radio channel 26, CCA threshol
d -45
Tentative link-local IPv6 address fe80:0000:0000:0000:0212:7401:0001:0101
Starting 'Border router process' 'Web server'
*** Address:fd00::1 => fd00:0000:0000:0000
Got configuration message of type P
Setting prefix fd00::
Server IPv6 addresses:
 fd00::212:7401:1:101
 fe80::212:7401:1:101
```

**Figure 8.** Border router connectivity with external network.

Finally, Copper (Cu) based CoAP user agent is installed for communication with web server which is then opened in browser for control and monitoring of IoT devices in smart agriculture. The Copper CoAP user agent is an add-on for the Firefox web browser, used for browsing and direct interaction with CoAP resources.

### 4.3. Operation and Procedure of Proposed CSFSA framework

In this phase, the operation of proposed CSFSA is presented in 3 primary modules such as the internal network i.e., IoT devices, border router and external network i.e., internet as illustrated in Figure 9. The framework's procedure and operation ratify the proposed IoT protocol stack which includes IoT devices that use CoAP at the application layer, a border router that uses 6LoWPAN at the network layer and IEEE 802.15.4 radio at the link layer and a web server that uses a Copper-based CoAP client. The RPL border router acts as bridge that connects external network with 6LoWPAN based internal network. The proposed framework connects CoAP based IoT devices to the internet via 6LoWPAN border router

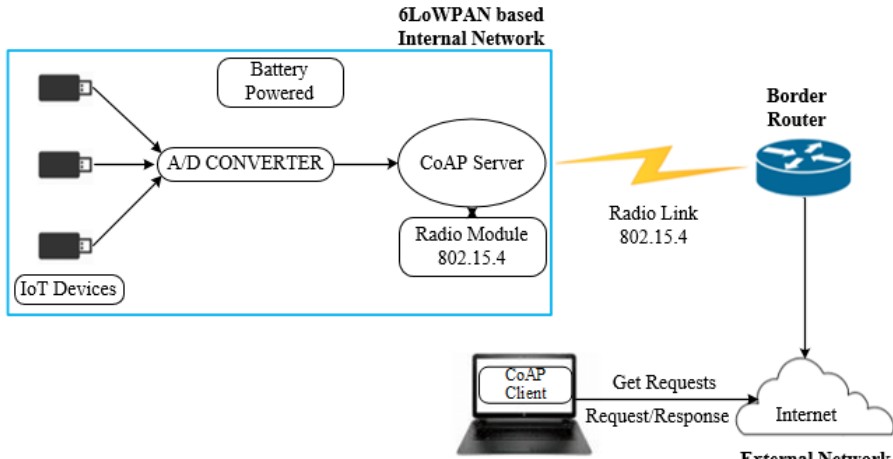

**Figure 9.** Operation of proposed CSFSA.

## 5. Results Evaluation and Discussion

To appraise the performance of proposed CSFSA, the CSFSA is emulated in Contiki NG operating system and Cooja simulator. Contiki NG is an open source operating system

which is designed especially for memory constrained systems. Cooja is an open source simulator which is used for the development of Contiki NG, as it provides simulation environment by allowing developers to simulate and test code/system before running it on the actual hardware.

*5.1. Simulation Parameters*

The simulation parameters used for the implementation of CSFSA is shown in Table 3. The mote chosen for simulation is Wismote which is MSP430 microcontroller based board having 16 KB of RAM and 256 KB of flash memory and a radio chip CC2420 compatible with IEEE 802.15.4 standard. The protocols used in the simulation are CoAP as an application layer protocol, UDP as transport layer protocol, 6LoWPAN as network layer protocol and IEEE 802.15.4 as link layer protocol.

**Table 3.** Simulation parameters.

| Parameter | Value |
| --- | --- |
| Operating System | Contiki-NG |
| Simulator | Cooja |
| Application Layer Protocol | CoAP |
| Network Layer Protocol | 6LoWPAN |
| Link Layer Protocol | IEEE 802.15.4 |
| MAC Layer Protocol | CSMA |
| Routing Protocol | RPL |
| Radio Propagation Model | UDGM Distance Loss |
| CoAP Client | Cupper based browser |
| Compiler | Msp430-gcc (in Ubuntu) |
| Sensor mote | Wismote |
| Number of motes | 5 |
| Data rate | 250 kbps |
| Tx/Rx Ratio | 100% |

*5.2. Scenario Development*

In order to evaluate the CSFSA's performance, it is divided into two simulation scenarios i.e., CoAP with proposed CSFSA and CoAP-DTLS implementation scenarios. CoAP with proposed CSFSA scenario comprised of 5 motes and 1 border router. Here motes are CoAP servers which are developed on proposed IoT protocol stack and deployed in agriculture field for control and monitoring of various agriculture parameters. The mote 1 is the border router while motes 2–6 running CoAP servers and all the motes are placed in the transmission range of border router. The border router is used for connectivity with external network (internet) through tunslip utility. The purpose of CoAP servers are monitoring and controlling of environmental conditions of agriculture field and send it to remote server i.e., CoAP client application. The CoAP client is Cupper based browser which is used for browsing and direct interaction with CoAP resources. The CoAP with proposed CSFSA scenario is emulated in Cooja as shown in Figure 10. While in CoAP-DTLS scenario, mote 1 is border router and motes 2–6 are CoAP servers and all the motes are placed in the transmission range of border router as shown in Figure 11. The CoAP-DTLS scenario is used for results comparison with proposed CSFSA framework based CoAP.

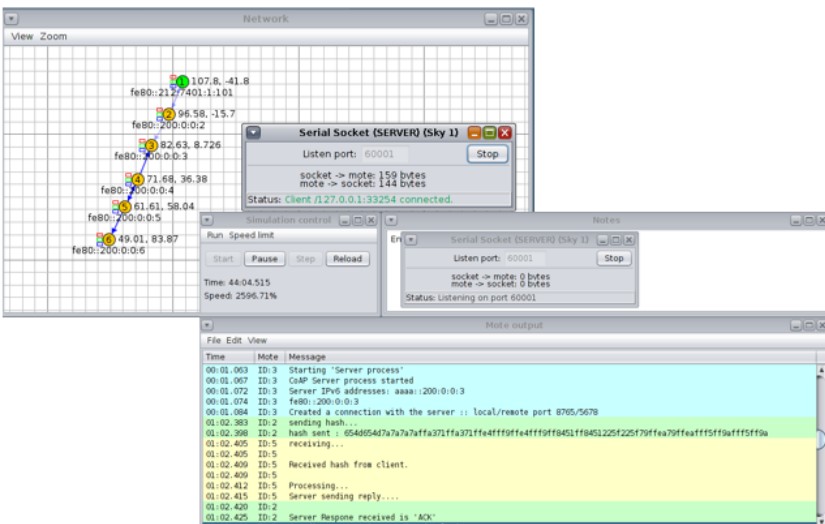

**Figure 10.** Proposed CSFSA simulation scenario consisting of 1 border router and 5 CoAP motes.

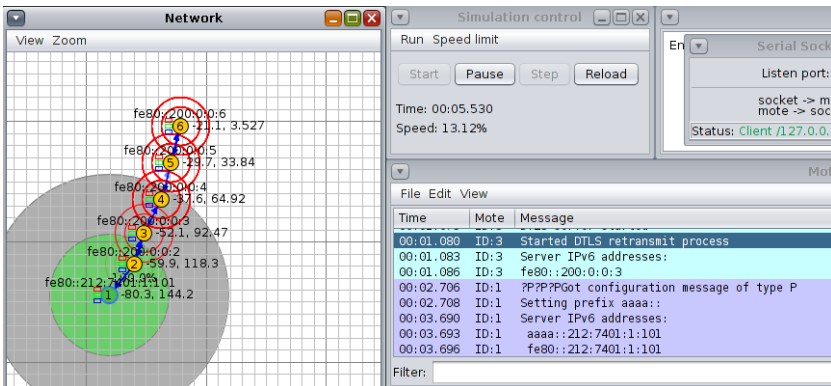

**Figure 11.** CoAP-DTLS simulation scenario consisting of 1 border router and 5 CoAP motes.

### 5.3. Results

This section describes the simulation results obtained. We appraised the results of proposed CSFSA in terms of performance evaluation parameters i.e., packet size, communication overhead and power consumption.

The packet size is an important evaluation parameter and have a significant impact on power consumption particularly in constrained networks. For message transmission, the standard CoAP requires one round trip. In proposed CSFSA framework, the authentication messages require two round trips as compared to CoAP-DTLS scheme, which processes it in four round trips i.e., three round trips for DTLS and one round trip for CoAP. Proposed CSFSA is lightweight and power efficient as compared to CoAP-DTLS and has significantly smaller packet size than CoAP-DTLS. Figure 12 illustrates the packet size and the number of messages for both proposed CSFSA framework and CoAP-DTLS scheme during transmission.

From the figure, it is clear that the packet size for our proposed CSFSA is smaller than that of packet size of CoAP-DTLS. Also the round trip time of our proposed CSFSA is lower as compared to round trip time of CoAP-DTLS.

Communication overhead is defined as the number of extra messages or packets that has been transmitted over network. The Standard CoAP communication overhead does not include extra packets for data transfer. The DTLS adds extra communication overhead to secure CoAP. To establish DTLS secure session, it adds 29 bytes to each datagram i.e., 8 bytes nonce and 8 bytes authentication tag. In CoAP-DTLS, DTLS handshake adds extra 29 bytes as overhead to standard CoAP. In our proposed CSFSA, it requires only two extra bytes communication overhead to the total number of transmitted bytes to secure the

communication between client and server. Figure 13 shows the communication overhead for both proposed CSFSA and CoAP-DTLS scheme. Based on comparison, communication overhead in our proposed CSFSA is lower as compared to CoAP-DTLS. Its means that our proposed CSFSA is efficient and has lightweight communication overhead.

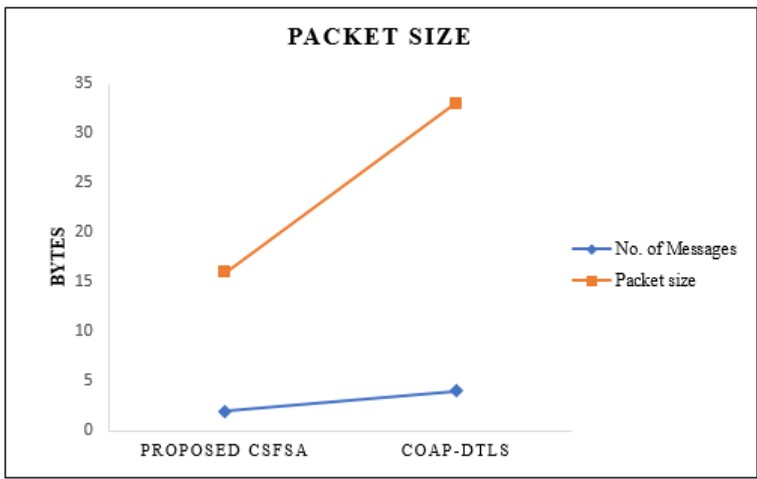

**Figure 12.** Packet size.

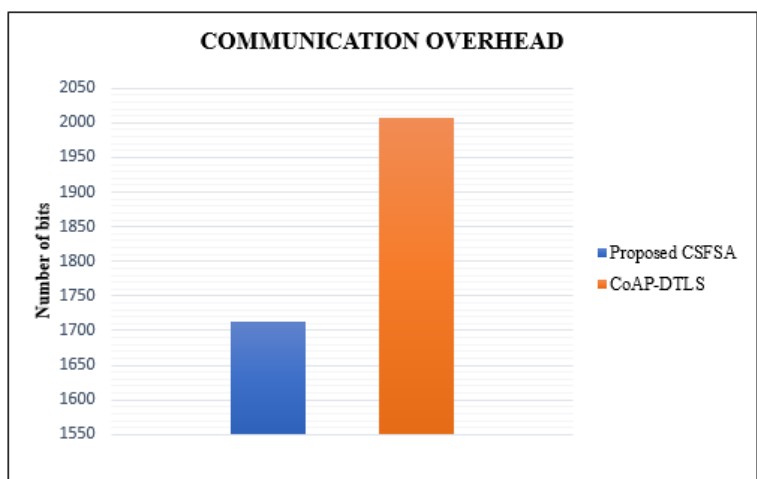

**Figure 13.** Communication overhead comparison.

Power consumption is another important metric, as it directly impacts the lifetime of IoT devices since IoT devices are powered by small sized batteries. Proposed CSFSA is lightweight and has significantly smaller packet size than CoAP-DTLS. Hence, it consumes less power and bandwidth as compared to CoAP-DTLS. The power consumption of proposed CSFSA is measured through software based energy estimation tool called Energest which is built in Contiki NG OS. The function energest_type_time () outputs the clock ticks obtained from the time when IoT device is booted. The formula for measuring the power consumption is as follows:

$$\text{Power Consumption} = \frac{\text{EnergestValue} \times \text{Current} \times \text{Voltage}}{\text{RTIMER\_ARCH\_SECOND} \times \text{RunTime}} \tag{8}$$

where current, voltage and RTIMER_ARCH_SECOND are constant, taken from Wismote datahseet. Figure 14 shows the power consumption measured for both proposed CSFSA and CoAP-DTLS scheme.

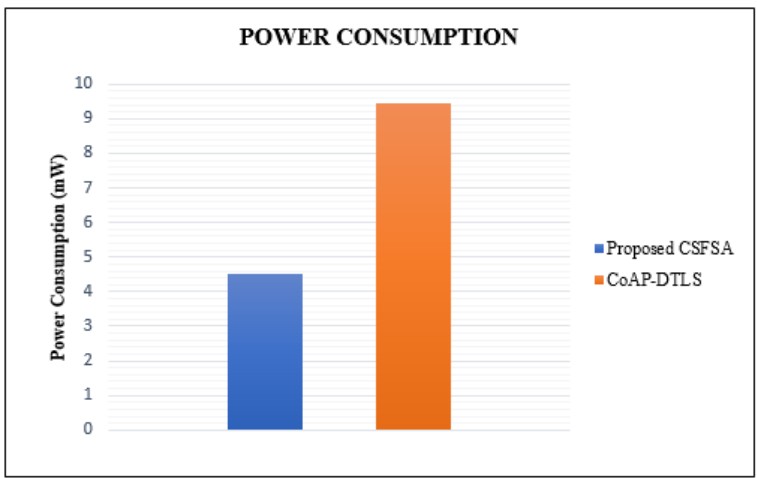

**Figure 14.** Power consumption.

## 6. Conclusions

Agriculture plays an important role in country's economic growth. In agriculture, the yields or crops which are to be produced are sensitive to changes in their environmental conditions and required controlled environment. Conventional agriculture methods are not integrated with IoT devices to provide controlled environment, resulting in low crop and fruit yields. This necessitates the integration of IoT technology in agriculture field to boost crop productivity. However, the integration of IoT in agriculture field introduces cyber security attacks and vulnerabilities. Such cyber attacks have the capability to adversely affect the countries' economies that are heavily reliant on agriculture. On the other hand, these IoT devices are resource constrained having limited memory and power capabilities and cannot be secured using conventional cyber security protocols. Therefore, designing robust and efficient secure framework for smart agriculture are required and is main aim of this paper.

In this paper, a Cyber Secured Framework for Smart Agriculture (CSFSA) is proposed. The proposed CSFSA presented a robust and tamper resistant authentication scheme for IoT devices using CoAP to ensure the data integrity and authenticity of smart agriculture monitoring. The significance of proposed CSFSA is to make resilient against various cyber security attacks i.e., replay attacks, Denial of Service (DoS) attacks, resource exhaustion and is efficient for constrained devices having limited memory. Furthermore, the packet size, communication overhead and energy consumption are also computed to appraise efficiency of proposed CSFSA framework. As a result, the proposed CSFSA can make a significant contribution to the problem of food wastage and its security. Also the economic loss can be reduced greatly.

**Author Contributions:** Conceptualization, W.U.K., S.A. (Saleh Alyahya) and S.N.K.M.; methodology, W.U.K. and S.A. (Salman Ahmed); software, W.U.K.; validation, W.U.K., S.A. (Salman Ahmed), S.N.K.M. and S.H.; formal analysis, W.U.K.; investigation, S.H.,W.U.K., S.A. (Salman Ahmed) and S.N.K.M.; resources,W.U.K., S.A. (Saleh Alyahya) and S.N.K.M.; data curation,W.U.K., S.A. (Saleh Alyahya) and S.H.; writing—original draft preparation, W.U.K.; writing—review and editing, S.A.,W.U.K., S.A. (Salman Ahmed), S.N.K.M. and S.H.; visualization, W.U.K.; supervision, S.A. (Salman Ahmed), and S.N.K.M.; project administration, S.A. (Salman Ahmed), S.A. (Saleh Alyahya), S.N.K.M. and S.H. All authors have read and agreed to the published version of the manuscript.

**Funding:** This research received no external funding.

**Acknowledgments:** This work is supported by the Department of Electrical Engineering, College of Engineering and Information Technology, Onaizah Colleges, Saudi Arabia.

**Conflicts of Interest:** The authors declare no conflict of interest.

## Abbreviations

The following abbreviations are used in this manuscript:

| | |
|---|---|
| IoT | Internet of Things |
| CSFSA | Cyber Secured Framework for Smart Agriculture |
| CoAP | Constrained Application Protocol |
| DTLS | Datagram Transport Layer Security Protocol |
| HTTP | Hyper Text Transfer Protocol |
| DoS | Denial of Service |
| AES | Advanced Encryption System |
| ECC | Elliptic Curve Cryptography |
| REST | Representational State Transfer |
| IETF | Internet Engineering Task Force |
| SSM | Secure Service Manager |
| TLS | Transport Layer Security |
| TCP | Transmission Control Protocol |
| UDP | User Datagram Protocol |
| IPv6 | IP version 6 |
| 6LoWPAN | IPv6 over Low-Power Wireless Personal Area Network |
| MQTT | Message Queuing Telemetry Transport |
| XMPP | Extensible Messaging and Presence Protocol |
| BLE | Bluetooth Low Energy |
| WSN | Wireless Sensor Network |
| IEEE | Institute of Electrical and Electronics Engineers |

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
