# Peer review of "Cyber Secure Framework for Smart Agriculture: Robust and Tamper-Resistant Authentication Scheme for IoT Devices"

_electronics, doi:10.3390/electronics11060963_

Round 1

Reviewer 1 Report

The paper has a good potential for being appreciated and cited, but it requires some improvements.

The section Introduction should clarify better and provide concise information with regard to the problem definition and scope of the paper. The contribution summarization should be remarked better. 

At the end of related works, highlight in some lines what overall technical gaps are observed in existing works, that led to the design of the proposed approach. Please consider the following works showing different approaches in this field: https://ieeexplore.ieee.org/abstract/document/9409962 https://www.sciencedirect.com/science/article/pii/S0950705121005037

Is it possible to consider another strategy to be compared?

The future scope of the methodology should be extended/highlighted. Improve the conclusion, clarify the conclusion of this article and its significance for follow-up research.

Minor comments:

Figures 2-5 have to be improved

Analysis about scalability features of the approach could be added to further improve the strength of the paper.

Author Response

Dear Reviewer 

Reviewer 2 Report

Overall the paper seems to be interesting and sound, however, in my opinion, it is still affected by some problems.

First of all, in some parts, the clarity and editorial quality of the paper weaken. As a consequence, such parts result to be quite difficult to read. Therefore, I would suggest to carefully improve the prose of writing in order to make this paper easier to read.

Furthermore, presentation aside, by reading the paper, it still was not entirely clear what to expect with the direction of the article. Indeed, the contribution proposed in this paper has been only marginally compared and contextualised with respect to the state of the art. As a result, it is extremely difficult to understand the novelty/contributions introduced by the paper. The aforementioned aspects should be carefully addressed before the paper can be considered any further.

The paper should be better compared and contextualized with respect to the state of the art. I want suggest these papers to authors:

- Lombardi, M., Pascale, F., and Santaniello, D. (December 14, 2021). "Two-Step Algorithm to Detect Cyber-Attack Over the Can-Bus: A Preliminary Case Study in Connected Vehicles." ASME. ASME J. Risk Uncertainty Part B. September 2022; 8(3): 031105. https://doi.org/10.1115/1.4052823

- Lombardi, M. , Pascale, F. , and Santaniello, D. , 2021, “ Internet of Things: A General Overview Between Architectures, Protocols and Applications,” Information, 12(2), p. 87.10.3390/info12020087

The figures should be better explained in their component parts

Finally, a thorough proofreading would be suggested, since in the paper there are some typos and formatting issues.

As remarks:

  • The paper should be better compared and contextualized with respect to the state of the art.
  • In some parts of the paper, the clarity and editorial quality of the paper weaken. As a consequence, such parts result to be quite difficult to read. Therefore, I would suggest to carefully improve the prose of writing in order to make this paper easier to read.
  • Each figure should be properly defined within the text and must be improved in quality.
  • An accurate proofreading is strongly recommended.

Author Response

Dear Reviewer

Reviewer 3 Report

The framework is novel and innovative. This would be considered as a contribution towards the domain of precision Agriculture Sector.

Author Response

Dear Reviewer

Round 2

Reviewer 1 Report

The authors addressed all my comments. In my opinion, the paper can be accepted.

Reviewer 2 Report

all suggested revisions have been applied. I recommend reviewing the paper for residual typos